# Attitude of the Lithuanian Public toward Medical Assistance in Dying: A Cross-Sectional Study

**DOI:** 10.3390/healthcare12060626

**Published:** 2024-03-10

**Authors:** Benedikt Bachmetjev, Artur Airapetian, Rolandas Zablockis

**Affiliations:** 1Faculty of Medicine, Vilnius University, M.K. Ciurlionio 21, LT-03101 Vilnius, Lithuania; artur.airapetian@mf.stud.vu.lt; 2Clinic of Chest Diseases, Immunology and Allergology, Institute of Clinical Medicine, Vilnius University, M.K. Ciurlionio 21, LT-03101 Vilnius, Lithuania; rolandas.zablockis@mf.vu.lt

**Keywords:** euthanasia, assisted suicide, human rights, terminal illness

## Abstract

Euthanasia and assisted suicide, involving the intentional termination of a patient’s life, are subjects of global debate influenced by cultural, ethical, and religious beliefs. This study explored the attitudes of the general public toward euthanasia, finding varying levels of support. A cross-sectional study was conducted. This research specifically evaluated the perspectives of 5804 Lithuanian residents using a survey distributed through social media, which presented medical scenarios on life-preserving interventions. Analysis indicated that gender, religion, experience in caring for patients in a terminal condition, education, and age significantly influenced the attitudes of the respondents toward end-of-life decisions. Specifically, factors like being non-religious or having less experience in caring for the terminally ill correlated with a more positive opinion regarding euthanasia and other forms of medical assistance in dying.

## 1. Introduction

Euthanasia and assisted suicide are controversial topics that are causing debates worldwide. Assisted suicide and active euthanasia are crucial yet distinct concepts in the discourse on end-of-life care. Assisted suicide is defined as the provision of means and information to a patient, enabling them to end their own life, typically in cases of severe, incurable disease [1]. In contrast, active euthanasia involves a healthcare professional directly administering a lethal injection to end a patient’s life at their request [2]. While some people view euthanasia as a soft end-of-life option, others consider it a violation of human rights and a type of murder.

Initially, in antiquity, perspectives on euthanasia were not universally negative, as evidenced by the texts of Aeschylus [3]. However, the Hippocratic Oath, a cornerstone of medical ethics, explicitly opposed it [4]. In the 20th and 21st centuries, the discourse around euthanasia has been reshaped by advancements in medicine, patient autonomy, and ethical considerations. This progression reflects a dynamic interplay between evolving societal values, medical practices, and legal systems, culminating in a diverse range of global perspectives on the issue.

Although 68% of patients in terminal stages report experiencing pain in their final week of life [5], managing their discomfort extends beyond merely prescribing opioids. Other measures are crucial to effectively alleviate their suffering. Contrary to common assumptions that uncontrollable pain is the predominant factor for assisted suicide, a report from Oregon indicates that the leading cause is a diminished ability to participate in enjoyable activities and the loss of dignity [6]. This necessitates a greater commitment to exploring methods that can maintain an individual’s activity and dignity for as long as possible. A subset of patients may develop tolerance to opioids [7], meaning that only a small percentage of them might face limited options for effective pain management. This finding underscores the intricate and multifaceted nature of the motivations behind such decisions, revealing that the factors influencing end-of-life choices extend beyond physical pain.

The general public’s opinion regarding end-of-life decisions varies across different countries, and it is influenced by a variety of factors. In some countries, euthanasia is legal and regulated, while in others, it is illegal and forbidden by law. Switzerland, for instance, has legalized assisted suicide exclusively [8], which is a decision that reflects a particular ethical and legal stance on autonomy and end-of-life care. Conversely, the Netherlands presents a broader legal framework, encompassing both assisted suicide and active euthanasia [9]. This dual legalization reflects a more inclusive approach to patient autonomy and decision making in terminal care. Furthermore, in Spain, euthanasia was legalized in 2021 [10], marking a significant shift in the country’s approach to end-of-life issues. The variation in legislation across different countries regarding assisted suicide and euthanasia highlights the controversial nature of this topic on a global scale.

As society ages and the accumulation of chronic illnesses increases [11], the issue becomes more significant. This trend signifies the growing importance of addressing these complex ethical matters in public discourse and policy making. Policy making heavily depends on the general public’s moral values, as they are the ones who vote in elections. These votes reflect societal norms and expectations, thereby influencing the direction and nature of policies formulated, especially in sensitive areas like end-of-life care.

The attitude toward euthanasia varies in different countries. According to data from the Pew Research Centre in 2013, 66% of Americans support the idea of allowing doctors to end the lives of patients in a terminal condition [12]. In the UK, where euthanasia is illegal, a different study found that 40% of the public supported the idea of adopting the law legalizing this practice [13]. However, in the USA, less than one-fifth of physicians report having heard a desire of the patients for medical assistance in dying [14]. However, in Croatia, more than one-third of the respondents (38.1%) agreed with terminating life-prolonging treatment for dying people experiencing intense and intolerable distress [15], while 71.2% opted in favor of active voluntary euthanasia in Austria [16]. Similarly, another study in Spain showed that 65% of the respondents were in favor of having the right to die from a critical illness [17].

The perception of euthanasia is shaped by various factors with religiosity being a significant influence. This religious influence might explain the absence of legalized euthanasia in Islamic nations [18]. Interestingly, research indicates that religiosity’s impact on attitudes toward euthanasia is more pronounced among medical professionals compared to those in other professions [19]. Perceptions of end-of-life care practices are surprisingly linked to several sociodemographic factors that might not be immediately obvious, including a nation’s Gross Domestic Product (GDP), average life expectancy, and rates of infant mortality [20].

It is worth mentioning that this exact topic has not been widely studied in Lithuania. That is why the factors influencing the attitude of the Lithuanian public toward this practice are still unknown. The sole study conducted in Lithuania on this topic was undertaken by S. Rakickaja for her Master’s Thesis in 2016 [21]. It is noteworthy, however, that the study’s sample size was limited, consisting of only 161 respondents, which raises concerns about its representativeness for broader population insights.

## 2. Materials and Methods

The objective of this study was to evaluate the perspectives of Lithuanian residents toward the termination of life for patients with a terminal illness. Data collection for the study took place over a one-month period, beginning on 21 December 2021, and concluding on 21 January 2022. This research included a group of individuals from different residences, gender, and age groups, and it had a study sample of 5804 respondents, 5790 of whom were suitable for the statistical analysis.

The participants were provided with information regarding the study’s objective and purpose. They were notified that their participation involved a simple acceptance click after reading this information, ensuring their personal data would be kept confidential and their identities irreversibly anonymous.

Selection Criteria:

Age: Participants had to be 18 years old or older.Language: Participants must be fluent in Lithuanian, as the survey was conducted in this language.Residency: Only residents of Lithuania were included in the study.

Exclusion Criteria:

Individuals below the age of 18.Non-Lithuanian speakers, to ensure comprehension and accurate responses, as the survey was exclusively in Lithuanian.Non-residents of Lithuania.Individuals who chose not to participate or did not give their consent for any reason, including moral or ethical concerns.

These criteria were designed to ensure a focused and relevant participant demographic for the study’s objectives.

The survey instrument was created using the Google Forms platform (as of December 2021) and was distributed through social media channels. Facebook, Twitter, LinkedIn, and Instagram were utilized in the study, enabling the collection of data from a wide demographic range, encompassing various ages, educational backgrounds, and other factors. This approach was crucial in ensuring a diverse and representative participant pool for the research. To ensure that our study participants were from Lithuania, several measures were implemented: the survey was conducted exclusively in Lithuanian, the description explicitly stated that it was focused on the attitudes of Lithuanians, and the survey was distributed in online groups where individuals engage in discussions about issues pertaining to Lithuania. These strategies collectively helped in targeting and obtaining responses from the intended demographic group.

The survey was designed to be anonymous. Participants had to indicate their consent by ticking a box before proceeding, ensuring they were aware their responses would be used anonymously for analysis and publication. This consent process upheld respondent anonymity while fully informing them about their participation in the study.

The survey included six medical scenarios exploring attitudes toward euthanasia and assisted suicide under various conditions:Bone Cancer in a Young Patient: The scenario involves a 25-year-old with advanced bone cancer, experiencing unrelenting pain and breathlessness, with a life expectancy of less than six months and only pain relief available, asking whether the patient should have the option to end his life using prescribed medication.Advanced Lung Cancer: This scenario describes a 42-year-old lung cancer patient in severe pain, with significant cancer progression and a prognosis of only three months, limited to pain management, asking whether the patient should have the option to end his life by requesting euthanasia.No Resuscitation Decision: A case of a 62-year-old patient with small cell lung cancer, whose condition deteriorates rapidly, leading to a heart stoppage shortly after hospitalization. The scenario questions the ethics of medical staff deciding not to resuscitate the patient without family consent.Living Will for a Healthy Individual: This scenario explores the idea of a healthy individual making an advance directive (living will) for non-application of life-saving measures in case of severe health deterioration.Chronic Mental Illness: Involves a 28-year-old patient suffering from long-term psychosis and treatment-resistant depression, with persistent psychological suffering, asking whether they should have the option to end their life using prescribed medication.

In each case, there was an option for respondents to agree or disagree with the discontinuation of medical treatments or with the termination of the patient’s life. The clinical scenarios presented in the survey were reviewed and validated by medical specialists, including a pulmonologist, an anesthesiologist, and a hematologist, to ensure their accuracy and relevance to the field. Moreover, the questionnaire contained demographic characteristics including age, gender, years of taking care of patients with a terminal condition, place of residence, and religious beliefs. These sociodemographic questions were selected based on precedents set by similar studies in this field. This approach ensured that the data collected were relevant and comparable to existing research, allowing for a more comprehensive understanding of the subject matter. The factors included in the questions of our data collection form are considered to be among the most significant in influencing attitudes toward end-of-life decisions.

Before its distribution, the questionnaire underwent a reassessment for face validity, which involves judging whether the questions seem appropriate and relevant for the intended construct and objectives. Given the clear and linguistically adapted nature of the questions to the Lithuanian context, they were found to be understandable, pertinent, and suitably concise. Since each item was evaluated on its own without contributing to a cumulative score, this level of validation was considered adequate for confirming the questionnaire’s appropriateness. Respondent had no time constraints for completion and were not offered incentives for participating.

All responses from the survey were exported to Excel and converted into numerical values. Descriptive and analytical statistical methods were used to analyze the data. A confidence interval of 95% was taken for prevalence values. Logistic regression models were made using SPSS version 26.0 (IBM Corp, Armonk, NY, USA). Fisher’s exact test and Pearson’s chi-squared test were used for a more detailed data analysis. Statistical significance was defined as *p* < 0.05.

Under Lithuanian law, our anonymous online survey did not require ethical approval as it did not fall within the legal parameters of a biomedical study.

## 3. Results

### 3.1. Demographic Characteristics of the Participants

Analyzing the data presented in Table 1, we can see the characteristics of the surveyed population:

The majority of survey participants were female, comprising 79.8% (*n* = 4622) of all participants, while only 20.2% (*n* = 1168) identified as male. The urban population was over-represented in the sample, as 90.8% (*n* = 5260) of the respondents live in urban areas compared to 9.2% (*n* = 530) in rural areas. In addition, 35.0% (*n* = 2024) of the respondents identified as non-religious and 65.0% (*n* = 3766) identified as religious.

Regarding the respondents’ experience in caring for patients with a terminal condition, the majority, 57.1% (*n* = 3307) of the sample, had no experience; 25.3% (*n* = 1467) had less than 1 year of experience; 10.6% (*n* = 615) had 1 to 5 years of experience; and only 6.9% (*n* = 401) had greater than 5 years of experience. For their educational background, most of the respondents, specifically 85.6% (*n* = 4958), had received higher education, while the remaining 14.4% (*n* = 832) had not.

Top of FormBottom of Form

The age distribution of the study can be summarized as follows: most respondents (49.9%; *n* = 2888) were younger than 35 years, while 24.9% (*n* = 1441) were aged between 35 and 45. Additionally, 16.0% (*n* = 932) of the participants were between 46 and 55 years of age, while only 9.1% (*n* = 529) were aged over 55 years old.

**Table 1 healthcare-12-00626-t001:** The demographical features of the respondents.

Demographic Variables		Number (Percent) of Respondents
Gender	Male	1168 (20.2)
	Female	4622 (79.8)
Age of respondents	<35 years	2888 (49.9)
	35–45 years	1441 (24.9)
	46–55 years	932 (16.0)
	>55 years	529 (9.1)
Place of residence	Rural	530 (9.2)
	Urban	5260 (90.8)
Religion	Religious	2024 (35.0)
	Non-religious	3766 (65.0)
Experience in caring of patients with a terminal condition	No experience	3307 (57.1)
	Less than 1 year	1467 (25.3)
	1 year to 5 years	615 (10.6)
	Over 5 years	401 (6.9)
Educational background	With higher education	4958 (85.6)
	No higher education	832 (14.4)

Data are shown as number (%).

### 3.2. The First Two Clinical Cases: Life Termination of Patients in a Terminal Condition Performed by Doctor or by the Patient

The dataset, shown in Table 2, represents the attitude of the Lithuanian respondents toward assisted suicide, predicted on various demographic parameters, including gender, religiosity, experience in caring for individuals with terminal illnesses, educational background, and age.

Seeing gender as a variable, most participants (both male and female) responded positively to the option of giving a terminal patient the right to terminate his life: 72.1% (*n* = 842) and 68.8% (*n* = 3179), respectively. Nevertheless, the chi-square test confirms that gender significantly impacts opinions about assisted suicide with statistical significance (*p* = 0.012).

The age of the participants also affects their attitude toward assisted suicide. Most of all age groups answered “yes” with the highest percentage among respondents under 35 years old (82.4%; *n* = 2381) and the lowest percentage among respondents between 46 and 55 years old (48.9%; *n* = 456). The chi-square test shows that the difference between the age groups is statistically significant (*p* < 0.001).

Most religious participants expressed a positive attitude toward assisted suicide, with 61% (*n* = 2298) of them accepting it, in contrast to only 30.4% (*n* = 1146) giving a negative response. Statistical analysis reveals a significant difference in attitude toward assisted suicide among respondents based on their religious affiliation (*p* < 0.001). Non-religious participants show stronger support for assisted suicide with 85.1% (*n* = 1723) of them answering affirmatively.

The experience in caring for patients in a terminal phase also has a big influence on the attitude toward assisted suicide. Participants who had no experience in caring for patients with a terminal illness were more likely to agree to assisted suicide with 78.9% (*n* = 2609) of them supporting it. This percentage is growing with increased years of experience, with only 52.1% (*n* = 209) of respondents with over 5 years of experience supporting assisted suicide. Statistical analysis confirms a big difference in opinion on assisted suicide in various groups (*p* < 0.001).

Finally, education is also considered a factor in the attitude toward assisted suicide with participants with higher education showing greater support for it. Among participants with advanced academic backgrounds, 67.6% (*n* = 3352) of them accepted it, while 24.4% (*n* = 1209) did not. Statistical analysis shows a big difference between individuals with elevated educational qualification and those without it in their attitude toward assisted suicide (*p* < 0.001).

The second dataset (Table 3) shows the attitude of the Lithuanian respondents toward active euthanasia. It presents the distribution of answers in the second clinical case according to different demographic variables, such as gender, religion, experience in caring for patients with a terminal condition, educational background, and age.

In terms of gender, a higher percentage of males (73.5%; *n* = 585) than females (70.1%; *n* = 3242) had a positive opinion of active euthanasia with a statistically significant difference (*p* = 0.013).

Age was also found to have a significant impact on attitudes toward active euthanasia. Respondents under 35 years of age were more inclined to favor active euthanasia (83.8%; *n*= 2421) compared to those aged 35–45 years (65.1%; *n* = 938), 46–55 years (50.5%; *n* = 471), and over 55 years (51.0%; *n* = 270), with a highly significant difference (*p* < 0.001).

In terms of religion, a higher percentage of non-religious respondents (85.9%; *n* = 1739) than religious respondents (62.7%; *n* = 2361) showed their acceptance for active euthanasia with a highly significant difference (*p* < 0.001).

Respondents with experience in taking care of patients in a terminal condition also showed a significant difference in their attitudes toward active euthanasia. Respondents who had no experience caring for patients with a terminal illness are more likely to support active euthanasia (80.5%; *n* = 2661) compared to those with less than 1 year (61.6%; *n* = 904), 1–5 years (54.5%; *n* = 335), and over 5 years (49.9%; *n* = 200) of experience with a highly significant difference (*p* < 0.001).

Educational background was also found to be a significant factor. Participants with advanced academic backgrounds were less inclined to support active euthanasia (69.0%; *n* = 3419) compared to those with no higher education (81.9%; *n* = 681) with a highly significant difference (*p* < 0.001).

### 3.3. Living Will Order and Not Resuscitating Patients in a Terminal Condition without Their Permission

The next dataset (Table 4) provides information on the attitude of the Lithuanian respondents toward not resuscitating patients in a terminal condition without their consent, while the other dataset (Table 5) provides information on the use of living wills. The demographic variables included in both datasets are consistent with those used in Table 2 and Table 3.

About 43% of female and 45% of male respondents were in favor of not resuscitating terminally ill patients without their consent with no significant gender difference. However, religious beliefs significantly influenced responses: 42% of religious and 47% of non-religious participants agreed to this. Age also played a role, with the highest agreement (49%) among those over 55 and the lowest (41%) among those under 35, showing a significant age-related difference.

Respondents with no experience caring for terminally ill patients were more likely to agree to non-consensual resuscitation (43.9%) compared to those with 1–5 years of experience (40.8%), showing significant differences by experience level. Urban and rural responses were similar with a slightly higher agreement in rural areas (48.1% vs. 43.3%) but not significantly so. Educational background did not show a significant difference in responses with both higher educated and less educated groups showing similar levels of agreement.

In the living wills data, 61.7% of males (721 respondents) supported it, slightly more than the 60.8% of females (2812 respondents), with a notable statistical difference. Age-wise, the highest support came from under 35-year-olds (66.6%, 1922 respondents), decreasing with age. Non-religious respondents showed significantly higher support (72.9%, 1476 respondents) compared to religious ones (54.6%, 2057 respondents).

Experience in caring for terminally ill patients significantly influenced attitudes toward living wills. Those with no such experience were more supportive (65.3%, 2161 respondents) compared to individuals with varying lengths of experience, showing less support (56.2–54.1%). Educational background also played a role with less educated respondents showing more support (68.9%, 573 respondents) than those with higher education (59.7%, 2960 respondents), indicating a substantial difference in attitudes based on educational levels.

When we compare the two datasets, we can see some differences in the attitudes toward not resuscitating patients in a terminal condition without their consent and the acceptance of a living will. While the gender difference in attitudes toward not resuscitating patients in a terminal condition without their consent is not significant, there is a significant difference between genders in terms of agreeing to the existence of a living will. All the other factors potentially influencing the attitude of the respondents are similar.

### 3.4. Assisted Suicide Based on Drug-Resistant Mental Disorders

Table 6 shows the distribution of answers in the 5th clinical case according to different characteristics of respondents. The dataset represents how people in different groups accept assisted suicide based on drug-resistant mental disorders.

Gender significantly influenced responses to assisted suicide in the 5th clinical case with more males answering “Yes” and females more often saying “No” or being undecided. Age also played a crucial role with younger respondents showing less favorability toward “Yes”. Religious beliefs were another key factor, with non-religious individuals more likely to answer “Yes” compared to religious respondents, who showed a higher tendency to say “No” or remain undecided. These findings indicate notable differences in attitudes toward assisted suicide based on gender, age, and religious affiliation.

Experience in caring for terminally ill patients and educational background significantly influenced responses to the 5th clinical case about assisted suicide. Those with no experience and less education were more likely to say “Yes”, while those with more experience or higher education were inclined to answer “No” or remain undecided. These differences indicate how personal and professional experiences, as well as educational levels, shape attitudes toward end-of-life care decisions.

In conclusion, the results of the study suggest that acceptance of assisted suicide for drug-resistant mental disorders is significantly associated with several demographic factors, including gender, religion, experience in caring for patients with a terminal condition, educational background, and age. These findings have important implications for healthcare providers and policymakers who are involved in taking care of patients with mental illness.

## 4. Discussion

Our results provide an understanding of the attitude of respondents from Lithuania toward various questions on end-of-life decisions, including assisted suicide, active euthanasia, non-resuscitation without consent, and the acceptance of living will. Some of the studies mentioned earlier are confirmed by our findings, while others are contradictory. The main finding of the study suggests that in Lithuania, respondents’ attitudes toward end-of-life decisions vary based on factors like gender, religious beliefs, experience with patients in a terminal condition, education level, and age. Notably, acceptance for assisted suicide concerning drug-resistant mental disorders is lower than for somatic disorders but influenced by similar factors.

The public attitude toward euthanasia is influenced by many factors. Based on research carried out in Spain, there are no differences in attitudes toward euthanasia according to the gender or age of the participants, which does not correspond to our data. However, religion is seen as an important factor influencing the attitude [22]. Moreover, according to some studies, sociodemographic factors such as religiosity play a greater role in attitudes toward euthanasia than disease severity [23]. Religion is thought to be a significant factor not just in the general population but among doctors as well [24]. The role of religiosity is obvious when looking at our analysis as well. Studies suggest that respondents with a worse academic background tend to support euthanasia more compared to those with higher education [25], which also corresponds to our findings. Additionally, political orientation may have an impact on the public’s attitude [22]. A unique study from South Korea identified poor physical status and comorbidity as prognostic factors for acceptance of medical assistance in dying [26]. It is necessary to point out that the attitude can be influenced even by economic status [20] and education [27]. One study analyzing the results from 33 European countries suggests that higher education positively affects acceptance of euthanasia. Of course, not only sociodemographic factors affect the attitudes. The circumstances and the status of the patient are also found to have a significant importance [28]. Our findings indicate that experience in caring for terminally ill patients significantly impacts attitudes, which appears to contrast with some results from other studies in this area [29].

The data show that most respondents stayed in favor of assisted suicide across most demographic variables. Gender had a slight preference for males and was a statistically significant variable. The role of religiosity in acceptance is not surprising. Religious beliefs often influence peoples’ moral and ethical values, and this is shown in the significantly higher acceptance rates amongst non-religious respondents. However, it should be noted that religiosity surprisingly does not always play a significant role, as evidenced by countries like Estonia and the Czech Republic [30]. This suggests that the influence of religiosity is not merely a matter of its presence but also involves the intensity of belief and the surrounding cultural context. Experience in caring for the patients in a terminal condition also had a clear influence. Those respondents who had experience with terminal patients had a more empathetic opinion, leading to their conservative attitudes toward end-of-life decisions.

The attitudes toward active euthanasia were similar to the trends regarding assisted suicide. Again, gender and religiosity played significant roles. One interesting observation is the higher acceptance among those without higher education. This can prove the influence of education in forming the moral and ethical values of people. It may show the importance of including bioethical topics in higher education programs.

The acceptance of not resuscitating patients in a terminal condition without their consent and the legalization of living will bring out some differences. Gender differences were not significant for non-resuscitation decisions, suggesting that in terms of more passive end-of-life decisions, gender does not play a role. The significant differences in attitudes based on religiosity are obvious, once again showing its role in end-of-life decisions. While there are some similarities between the two situations, such as the significant differences in attitudes based on religious affiliation, experience in caring for patients with a terminal condition, and education, there are also some differences in attitudes toward not resuscitating patients in a terminal condition without their consent and the existence of a living will. These differences suggest that the two issues are not necessarily interchangeable and may require different approaches and wider studies of the topic.

Our findings indicate that attitudes toward assisted suicide in Lithuania vary significantly depending on whether the illness is somatic or mental. This suggests a distinct societal perception of these two types of diseases in the context of end-of-life decisions. Younger individuals seemed to be more accepting of it; however, the significant differences based on education, religion, and caregiving experience show that the same sociodemographic factors influence the attitude of respondents even in this situation. However, the level of acceptance observed in our data is notably lower when compared to the figures from the Netherlands [31].

The attitude of the Lithuanian public toward euthanasia can be more comprehensively understood by examining it from three key perspectives:Religious Influence in Lithuania: Christianity’s prevalence in Lithuania significantly shapes public opinion on euthanasia. Religious doctrines, emphasizing the sanctity of life, often lead to conservative viewpoints on such ethical issues. The study’s results reflect this influence with religious respondents showing lower acceptance of euthanasia and assisted suicide. This trend suggests a strong correlation between religious beliefs and attitudes toward end-of-life decisions, highlighting the need for better handling of these issues in religious societies.Historical Context and Older Population: The conservative outlook of Lithuania’s older population can be partly attributed to its Soviet past. Those raised in the USSR era were exposed to different societal and moral frameworks, which may influence their perspectives on contemporary ethical dilemmas like euthanasia. This generational divide is evident in our findings, where older respondents exhibited more conservative attitudes, underlining the impact of historical and cultural contexts on public opinion.Impact of Caregiving Experience: Individuals with experience in caring for terminally ill patients often have a negative perspective on euthanasia. Our study suggests that this experience tends to result in less favorable opinions toward euthanasia. This could be due to the observation that many terminally ill patients may not desire euthanasia.

However, the changing political landscape in Lithuania, particularly with the formation of a liberal-leaning coalition in 2020, suggests a shift in societal attitudes toward more liberal views on issues like euthanasia. This political shift is mirrored in the gradual change in public opinion, as observed in our study, where attitudes toward euthanasia and assisted suicide are more favorable in younger generation. This evolution reflects the influence of political ideologies and government policies on societal attitudes toward complex ethical issues.

This study offers an exploration into Lithuanian attitudes toward end-of-life decisions, set within the country’s unique context as the only Catholic nation in the former Soviet Union. This distinctive backdrop enriches the study, linking public opinions to a historical intersection of Catholic and Soviet influences.

The robustness of this study is further enhanced by its notably large sample size of 5790 respondents, which is substantially larger than those in similar studies conducted by entities such as the Pew Research Center (1994 respondents) [12] and in countries like Croatia (1203 respondents) [15] and Austria (1000 respondents) [16]. This expansive sample allows for an in-depth exploration of diverse viewpoints across various segments of the Lithuanian society. Importantly, it identifies specific groups more resistant to end-of-life options, providing a foundation for targeted discussions and interventions in Lithuania to address their concerns and viewpoints.

Moreover, the study provides a unique overview of the relationship between experiences in caring for terminally ill patients and attitudes toward end-of-life decisions. This aspect, not commonly observed in other studies, adds a critical dimension to understanding how personal involvement in end-of-life care shapes perceptions and choices in this sensitive area.

## 5. Conclusions

Our findings reveal that variables like gender, age, religious beliefs, caregiving experience, and educational background have a significant impact on Lithuanian attitudes toward assisted suicide and active euthanasia. For example, younger respondents and non-religious individuals exhibited a stronger inclination toward assisted suicide. There were gender differences in attitudes toward active euthanasia with a larger proportion of males showing support. Regarding active euthanasia, our study highlighted gender differences; more male respondents supported it compared to females. This trend was especially prominent among younger participants, particularly those under the age of 35, who demonstrated the highest acceptance. Furthermore, non-religious respondents displayed a more favorable stance toward active euthanasia than their religious counterparts. In terms of living wills, a similar trend was observed with younger and non-religious individuals showing greater support.

It is crucial to understand that the process of making decisions regarding end-of-life care, especially for those with advanced and progressive chronic illnesses, requires a deep understanding, careful reflection, and high sensitivity. This is not only a medical issue but a complex combination of ethical, societal, and economic factors. These decisions have far-reaching implications, influencing both the allocation of financial and governmental resources and the broader societal understanding of the dying process. The ethical complexity surrounding end-of-life decisions, particularly in the context of euthanasia and assisted suicide, is immense. These decisions often involve deep moral questions about the sanctity of life, autonomy, the role of suffering, and the responsibilities of medical professionals. Ethical debates focus on reconciling respect for patient autonomy with the ethical principle of ‘do no harm’. The diverse cultural, religious, and personal beliefs about death and dying further complicate these discussions, making it a deeply nuanced and challenging area of medical ethics. Recognizing the complex nature of these decisions is crucial in developing policies and practices that are empathetic, informed, and aligned with the moral values of the society.

## 6. Limitations of the Study

Sample: Given the over-representation of urban residents and those with higher education, our findings might not accurately reflect the views of the entire Lithuanian population.

Self-Selection: The use of online platforms for survey distribution may lead to self-selection bias, as only those who are active on these platforms and interested in the topic are likely to respond.

Cross-Sectional Design: The study’s cross-sectional nature limits the ability to infer causality or changes in attitudes over time.

Cultural and Social Influences: The study is specific to Lithuania and may not account for cultural and social influences that could affect attitudes differently in other regions.

Language Barrier: The survey being in Lithuanian might have excluded non-Lithuanian speaking residents, affecting the diversity of responses.

## Figures and Tables

**Table 2 healthcare-12-00626-t002:** The distribution of answers in the first clinical case according to different characteristics of respondents.

Demographic Variables		Number (Percentage) of Respondents			Significance		
		“Yes”	“No”	“Can’t Decide”	X^2^	df	*p*-Value
Gender	Male	842 (72.1%)	259 (22.2%)	67 (5.7 %)	8.820	2	**0.012**
	Female	3179 (68.8%)	1067 (23.1%)	376 (8.1%)			
Age of respondents	<35 years	2381 (82.4%)	343 (11.9%)	164 (5.7%)	548.723	6	**<0.001**
	35–45 years	915 (63.5%)	396 (27.5%)	130 (9.0%)			
	46–55 years	456 (48.9%)	382 (41.0%)	94 (10.1%)			
	>55 years	269 (50.9%)	205 (38.8%)	55 (10.4%)			
Religion	Religious	2298 (61.0%)	1146 (30.4%)	322 (8.6%)	388.195	2	**<0.001**
	Non-religious	1723 (85.1%)	180 (8.9%)	121 (6.0%)			
Experience in caring of patients in a terminal condition	No experience	2609 (78.9%)	467 (14.1%)	231 (7.0%)	379.944	6	**<0.001**
	Less than 1 year	872 (59.4%)	466 (31.8%)	129 (8.8%)			
	1 year to 5 years	331 (53.8%)	233 (37.9%)	51 (8.3%)			
	Over 5 years	209 (52.1%)	160 (39.9%)	32 (8.0%)			
Education	Higher	3352 (67.6%)	1209 (24.4%)	397 (8.0%)	55.676	2	**<0.001**
	Non-higher	669 (80.4%)	117 (14.1%)	46 (5.5%)			

Significant values are shown in bold.

**Table 3 healthcare-12-00626-t003:** The distribution of answers in the second clinical case according to different characteristics of respondents.

Demographic Variables		Number (Percentage) of Respondents			Significance		
		“Yes”	“No”	“Can’t Decide”	X^2^	df	*p*-Value
Gender	Male	858 (73.5%)	258 (22.1%)	52 (4.5%)	8.673	2	**0.013**
	Female	3242 (70.1%)	1078 (23.3%)	302 (6.5%)			
Age of respondents	<35 years	2421 (83.8%)	335 (11.6%)	132 (4.6%)	568.467	6	**<0.001**
	35–45 years	938 (65.1%)	401 (27.8%)	102 (7.1%)			
	46–55 years	471 (50.5%)	389 (41.7%)	72 (7.7%)			
	>55 years	270 (51.1%)	211 (39.9%)	48 (9.1%)			
Religion	Religious	2361 (62.7%)	1154 (30.6%)	251 (6.7%)	373.078	2	**<0.001**
	Non-religious	1739 (85.9%)	182 (9.0%)	103 (5.1%)			
Experience in caring of patients in a terminal condition	No experience	2661 (80.5%)	454 (13.7%)	192 (5.8%)	421.256	6	**<0.001**
	Less than 1 year	904 (61.6%)	479 (32.6%)	84 (5.7%)			
	1 year to 5 years	335 (54.5%)	232 (37.7%)	48 (7.8%)			
	Over 5 years	200 (49.9%)	171 (42.6%)	30 (7.5%)			
Education Background	Higher	3419 (69.0%)	1226 (24.7%)	313 (6.3%)	59.837	2	**<0.001**
	Non-higher	681 (81.9%)	110 (13.2%)	41 (4.9%)			

Significant values are shown in bold.

**Table 4 healthcare-12-00626-t004:** The distribution of answers in the third clinical case according to different characteristics of respondents.

Demographic Variables		Number (Percentage) of Respondents			Significance		
		“Yes”	“No”	“Can’t Decide”	X^2^	df	*p*-Value
Religion	Religious	1571 (41.7%)	1774 (47.1%)	421(11.2%)	42.377	2	**<0.001**
	Non-religious	959 (47.4%)	777 (38.4%)	288 (14.2%)			
Age of respondents	<35 years	1184 (41.0%)	1317 (45.6%)	387 (13.4%)	22.429	6	**0.001**
	35–45 years	654 (45.4%)	630 (43.7%)	157 (10.9%)			
	46–55 years	432 (46.4%)	388 (41.6%)	112 (12.0%)			
	>55 years	260 (49.2%)	216 (40.8%)	53 (10.0%)			
Experience in caring of patients with a terminal condition	No experience	1452 (43.9%)	1393 (42.1%)	462 (14.0%)	31.898	6	**<0.001**
	Less than 1 year	638 (43.5%)	674 (45.9%)	155 (10.6%)			
	1 year to 5 years	251 (40.8%)	300 (48.8%)	64 (10.4%)			
	Over 5 years	189 (47.1%)	184 (45.9%)	28 (7.0%)			

Significant values are shown in bold.

**Table 5 healthcare-12-00626-t005:** The distribution of answers in the fourth clinical case according to different characteristics of respondents.

Demographic Variables		Number (Percentage) of Respondents			Significance		
		“Yes”	“No”	“Can’t Decide”	X^2^	df	*p*-Value
Gender	Male	721 (61.7%)	342 (29.3%)	105 (9.0%)	9.230	2	**0.010**
	Female	2812 (60.8%)	1254 (27.1%)	556 (12.1%)			
Age of respondents	<35 years	1922 (66.6%)	643 (22.3%)	323 (11.2%)	105.697	6	**<0.001**
	35–45 years	824 (57.2%)	428 (29.7%)	189 (13.1%)			
	46–55 years	498 (53.4%)	342 (36.7%)	92 (9.9%)			
	>55 years	289 (54.6%)	183 (34.6%)	57 (10.8%)			
Place of residence	Rural	312 (58.9%)	149 (28.1%)	69 (13.0%)	1.819	2	0.403
	Urban	3221 (61.2%)	1447 (27.5%)	592 (11.3%)			
Religion	Religious	2057 (54.6%)	1265 (33.6%)	444 (11.8%)	215.492	2	**<0.001**
	Non-religious	1476 (72.9%)	331 (16.4%)	217 (10.7%)			
Experience in caring of patients with a terminal condition	No experience	2161 (65.3%)	737 (22.3%)	409 (12.4%)	116.791	6	**<0.001**
	Less than 1 year	824 (56.2%)	479 (32.7%)	164 (11.2%)			
	1 year to 5 years	331 (53.8%)	233 (37.9%)	51 (8.3%)			
	Over 5 years	217 (54.1%)	147 (36.7%)	37 (9.2%)			
Education	Higher	2960 (59.7%)	1401 (28.3%)	597 (12.0%)	27.622	2	**<0.001**
	Non-higher	573 (68.9%)	195 (23.4%)	64 (7.7%)			

Significant values are shown in bold.

**Table 6 healthcare-12-00626-t006:** The distribution of answers in the fifth clinical case according to different characteristics of respondents.

Demographic Variables		Number (Percentage) of Respondents			Significance		
		“Yes”	“No”	“Can’t Decide”	X^2^	df	*p*-Value
Gender	Male	470 (40.2%)	494 (42.3%)	204 (17.5%)	43.061	2	**<0.001**
	Female	1410 (30.5%)	2164 (46.8%)	1048 (22.7%)			
Age of respondents	<35 years	1178 (40.8%)	1117 (38.8%)	593 (20.5%)	214.119	6	**<0.001**
	35–45 years	382 (26.5%)	732 (50.8%)	327 (22.7%)			
	46–55 years	174 (18.7%)	547 (58.7%)	211 (22.6%)			
	>55 years	146 (27.6%)	262 (49.5%)	121 (22.9%)			
Religion	Religious	995 (26.4%)	2009 (53.4%)	762 (20.2%)	260.903	2	**<0.001**
	Non-religious	885 (43.7%)	649 (32.1%)	490 (24.2%)			
Experience in caring of patients with a terminal condition	No experience	1214 (36.7%)	1291 (39.0%)	802 (24.3%)	149.182	6	**<0.001**
	Less than 1 year	406 (27.7%)	789 (53.8%)	272 (18.5%)			
	1 year to 5 years	160 (26.0%)	346 (56.3%)	109 (17.7%)			
	Over 5 years	100 (24.9%)	232 (57.9%)	69 (17.2%)			
Education	Higher	1504 (30.3%)	2379 (48.0%)	1075 (21.7%)	80.893	2	**<0.001**
	Non-higher	376 (45.2%)	279 (33.5%)	177 (21.3%)			

Significant values are shown in bold.

## Data Availability

The dataset supporting the conclusions of this publication is proprietary and will not be publicly shared. However, additional information regarding the study methodology and analysis can be provided upon reasonable request to the corresponding author.

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
