# Peer review of "Attitude of the Lithuanian Public toward Medical Assistance in Dying: A Cross-Sectional Study"

_healthcare, 2024, doi:10.3390/healthcare12060626_

Round 1

Reviewer 1 Report

Comments and Suggestions for Authors

The document under analysis portrays a topic of interest to the scientific community, due to its relevance, topicality, and importance for political decision-making regarding the end of people's lives.

The introduction is a little poor. It needs more content, both in terms of the central concepts under analysis and the relevance of the topic for civil society. We consider it relevant that the authors objectively define assisted suicide and active euthanasia. Showing the position of global health regulatory bodies on the topic.

Making decisions that will determine the financial and governmental support that is aimed at supporting how people with advanced and progressive chronic illness can end their lives, requires a lot of knowledge, reflection and sensitivity regarding the topic "the process of dying". The authors need to support the introduction with a much more current bibliography and bibliography.

Poor method description. We do not have information about the selection/exclusion criteria of participants, we do not know the time window in which data collection takes place. We do not know the criteria used to construct the data collection form. We know very little about the constituents of this form. We suggest that they be referred to more objectively

The Results are difficult to read, and the information in the tables is repeated in the text. The tables could also have an organizational structure that facilitates understanding of the information by readers.

Reviewer 2 Report

Comments and Suggestions for Authors

Overall: It is an interesting piece of research and adds to the existing body of knowledge.

Title: Title of the research sounds good and words length is also acceptable.

Abstract: It is good. Keywords are mentioned.

Introduction: The introduction aligns with the objectives. The introduction is well written. However more conceptual clarity would have strengthened the paper. Is there any thin line difference between euthanasia and assisted suicide? A global historical perspective would give a better footing to the article. Review of related studies only limited to USA and European countries, what about other countries and continents?

Method & Design: The research approach used for this study to realise the objectives of the study is appropriate.

Who developed (field of expertise) the medical scenarios? What are the psychometric properties of the instrument used? Which social media platforms were used? What was the period the responses were collected? How did you ensure that all your respondents were from Lithuania? What about ethical approval of the study?

Use of Chi Squares to find the significant differences in the distribution is appropriate.

Results: All results have been reported correctly.

Discussion: The discussion section is well presented with appropriate refences.

Line no: 320-321 However, mental disorders, unlike terminal physical illnesses, are often seen differently in Lithuanian society. State the reference?

Conclusion: The conclusion has clarity, but it could be further strengthened.

Limitations of study: Outline the limitations of the current research.

All the following sections have been satisfactorily answered: Acknowledgments, Author Contributions, Conflicts of Interest, Funding, Data Availability Statement

Comments on the Quality of English Language

good

Reviewer 3 Report

Comments and Suggestions for Authors

First of all, congratulations for the work carried out.

Regarding the introduction, this is complete but more current references would be needed to provide current data, since there are references from 2013.

Considering the material and methods section, this is correctly explained but it would be necessary to emphasize more about whether the instrument is ad- hoc or validated and the different items included.

The results section is correct, interesting and well expressed.

Regarding the discussion, there is a lack of current studies with which to compare the results obtained.

The references section has some flaws, since not all references follow the regulations, in addition to there being studies that are too old.

Reviewer 4 Report

Comments and Suggestions for Authors

The article in question addresses a relevant and topical issue.

Despite the importance of the proposed approach, the article in its current form has several weaknesses that need to be corrected.

The introduction is very brief and insufficiently substantiated. In the first paragraph, it is necessary to explain in more detail the assumptions underlying the different positions that mark the debate on euthanasia and assisted suicide. If it is stated that the debate is controversial, a quick mention is not enough. It's important to make it clearer what the nature of the divides are (in order to better understand the characteristics and impact of the controversies) and it's also important to introduce some bibliographic support to better back up these arguments. The bibliography could be strengthened to cover this aspect. Although these are more general questions, they are fundamental for contextualising this topic.

In the case of the second paragraph of the introduction, it is not clear why there is not a single line of justification for the option of prioritising the study of the general public’s opinion on these issues. It is necessary to put forward arguments that show why this option is relevant, useful and necessary.

In the third and final paragraph, differences in the results of studies carried out in various countries are presented. However, nothing is said about the conclusions that could justify this diversity. The differences are merely pointed out, but are not the subject of any comment, reflection or discussion.

Regarding section 2, the nature of the sample presented is not clear. Is it statistically representative? Is it representative of the sociodemographic composition of the Lithuanian population? Is it stratified?

The second paragraph of this same section mentions the use of informed consent and the next paragraph states that the questionnaires were distributed through social media channels. To make the procedures of this methodological approach clear, it is necessary to explain the operational details. How and when was this application carried out? And considering that the answers were given online, it is to be assumed that they automatically become anonymous, so there is no point in using informed consent. This instrument presupposes the identification of the person who consents, but it also presupposes (as stipulated by the General Data Protection Regulation) that in order for the identity of the participants to be preserved from disclosure, the personal data is pseudo-anonymised by the researcher responsible. Was this the case? What exactly happened in this survey? If the answers are anonymous, the requirement for informed consent contradicts the assumption of anonymisation. It would be appropriate to use a participant information sheet that at no point requires a signature and consent, as this undermines the requirement of anonymity.

It would also be important to have more clarification on the content of the survey. Six medical scenarios are mentioned, but little is known about the actual content of the survey. What aspects were explored around these scenarios?

As for the content of the article, it is clear that section 3 is exclusively descriptive, limiting itself to characterising the distribution of responses according to the different variables. It is therefore essential to greatly strengthen and improve section 4, which is dedicated to discussing the results. It needs to be more developed so that it becomes more consistent in terms of the discussion itself. In fact, for the first sentence of the first paragraph to really make sense (“our results provide an understanding...”) you really need to invest in the discussion, as it adds little to the simple characterisation and description.

For a more interpretative approach to the data, it is necessary not only to articulate it with the sociodemographic characteristics of the Lithuanian population, but also to relate it to the sociocultural reality of the country in question, so that we can draw conclusions about the meaning of the trends presented, but also the reasons behind the different positions according to the variables presented.

Finally, it should be noted that the conclusion is excessively summarised and also inadequate. There is nothing substantively relevant in the conclusion. Nothing is said about the main contributions of this study. What does it add and what elements does it introduce for the development of the debate? What limitations does it have? What fundamental conclusions can be highlighted as truly significant?

In short, for the publication of this article to be viable, it needs to be revised and improved. In its current form, it is rather superficial and scientifically irrelevant. The description predominates and there is no relevant discussion to show that there is any effective addition or contribution to furthering knowledge in this field.

Round 2

Reviewer 2 Report

Comments and Suggestions for Authors

Revised version could be accepted.

Author Response

Thank you for reviewing the manuscript and providing valuable recommendations.

Reviewer 4 Report

Comments and Suggestions for Authors

The response from the authors, as well as the reformulations made to the text, are indicative of a successful effort to improve. The current version is frankly better than the previous one and manages to address the bulk of the questions raised.

At this stage, I would just like to share two observations:

1 1)  Regarding the issue of informed consent, I think it would be more appropriate to change that reference to participant information. When someone signs an informed consent form, it means that the researchers will be able to return to contact with the participants if they want or need to. In other words, this means that the information is not in itself anonymous. It is anonymised by the researchers (this is called pseudonymisation) in order to safeguard the identity of the participants, but the identity is not unknown to the researcher. In this case, if completing the questionnaire by agreeing to take part makes it impossible to trace the participant's identity, then what is required is not informed consent (which implies a signature on the part of the participants), but a simple click on the acceptance form after reading the information content. It's an apparently discreet difference, but we're talking about different things. If in this study the questionnaire was filled in irreversibly anonymously, then the right thing to do is to replace the references to informed consent with information for the participants.

      2) The paragraph that has been included between lines 438 and 452 is relevant, but it seems somewhat out of place. In terms of coherence and reading fluidity, it would make more sense to appear at the beginning of the conclusion.

Author Response

Thank you for your clarification on the issue of informed consent. We completely concur with your perspective and have accordingly updated the text:

1.1 Changing the "informed consent" term to a sentence:

"They were notified that their participation involved a simple acceptance click after reading this information, ensuring their personal data would be kept confidential and their identities irreversibly anonymous." (lines 102-104)

1.2 We have replaced the previous reference to obtaining informed consent with a more precise participant information statement:  "All subjects involved in the study were provided with detailed information about the study and indicated their willingness to participate through a simple acceptance click as a form of participant information statement." (lines 544-546)   2. We have also relocated the paragraph you pointed out to the conclusion section, ensuring a more cohesive and fluid reading experience. (lines 505-519)

For your convenience, all modifications have been made in a review mode.

Thank you once again for your valuable suggestions.